# Video-R1: Reinforcing Video Reasoning in MLLMs

**Kaituo Feng**[1], **Kaixiong Gong**[1], **Bohao Li**[2], **Zonghao Guo**[3*], **Yibing Wang**[4],
**Tianshuo Peng**[1], **Junfei Wu**[4], **Xiaoying Zhang**[5], **Benyou Wang**[2], **Xiangyu Yue**[1*]

[1]CUHK MMLab, [2]CUHK (SZ), [3]Tsinghua University, [4]UCAS, [5]CUHK HCCL

https://github.com/tulerfeng/Video-R1

## Abstract

Inspired by DeepSeek-R1's success in eliciting reasoning abilities through rule-based reinforcement learning (RL), we introduce Video-R1 as the first attempt to systematically explore the R1 paradigm for incentivizing video reasoning within multimodal large language models (MLLMs). However, directly applying RL training with the GRPO algorithm to video reasoning presents two primary challenges: (i) a lack of temporal modeling for video reasoning, and (ii) the scarcity of high-quality video-reasoning data. To address these issues, we first propose the T-GRPO algorithm, which encourages models to utilize temporal information in videos for reasoning. Additionally, instead of relying solely on video data, we incorporate high-quality image-reasoning data into the training process. We have constructed two datasets: Video-R1-CoT-165k for SFT cold start and Video-R1-260k for RL training, both comprising image and video data. Experimental results demonstrate that Video-R1 achieves significant improvements on video reasoning benchmarks such as VideoMMMU and VSI-Bench, as well as on general video benchmarks including MVBench and TempCompass, etc. Notably, Video-R1-7B attains a 37.1% accuracy on video spatial reasoning benchmark VSI-bench, surpassing the commercial proprietary model GPT-4o. All code, models, and data are released in https://github.com/tulerfeng/Video-R1.

## 1 Introduction

Recent advancements in rule-based Reinforcement Learning (RL) [17] have significantly enhanced the reasoning capabilities of Large Language Models (LLMs) [16, 11]. In particular, DeepSeek-R1 [11] has demonstrated that carefully designed RL pipelines can lead to emergent and robust reasoning abilities with long chain-of-thoughts (CoT) in text-based domains. Motivated by this success, several recent efforts have explored extending RL training to Multimodal Large Language Models (MLLMs) [33, 14, 39]. Notable examples include Kimi k1.5 [33] and Skywork R1V [39], which apply RL to improve reasoning over image-text pairs. However, despite these early explorations, the domain of video reasoning in MLLMs remains underexplored.

To bridge this gap, we present Video-R1 model, as the first attempt to systematically investigate eliciting strong video reasoning based on the R1 paradigm . However, directly applying RL training with the Group Relative Policy Optimization (GRPO) algorithm [30] to video reasoning introduces two fundamental challenges: First, original GRPO lacks explicit reward signals for encouraging temporal reasoning in a video. Without explicit temporal awareness, the model may take shortcuts for reasoning, focusing on a single frame or snapshot rather than reasoning over time (see Figure 1 for example; Video-UTR [40] also identifies a similar issue). The underexploitation for temporal cues can cause the learned reasoning strategies to "shortcut" the process—relying on superficial visual patterns, rather than engaging in deeper and temporally grounded reasoning. This could ultimately hindering generalization to more complex or diverse video reasoning tasks.

---

*Corresponding Authors

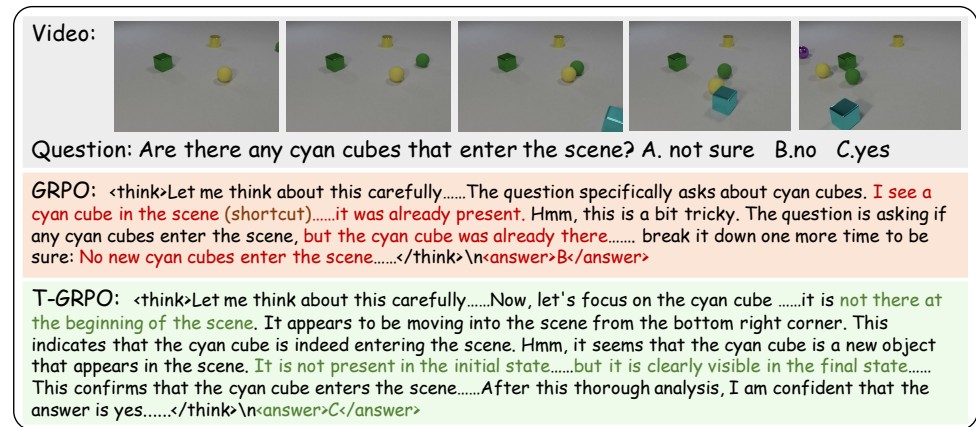

Figure 1: Reasoning paths of Video-R1 trained by GRPO and our proposed T-GRPO on test samples. Without explicit temporal modeling, models may learn sub-optimal video reasoning patterns by taking shortcuts, therefore failing to generalize well.

The second issue lies in the scarcity of high-quality video reasoning training data, especially samples that demand strong reasoning ability or involve long reasoning path. Most existing video training datasets mainly focus on simple recognition tasks, rather than reasoning. This scarcity makes it difficult to expose the model to diverse, challenging reasoning patterns during training, limiting the effectiveness of RL and hindering the emergence of robust reasoning behaviors.

To address these challenges, we propose two key solutions. First, we propose T-GRPO, an extension of the original GRPO algorithm [30] that explicitly encourages temporal reasoning. During training, the model is presented with both temporally ordered and randomly shuffled frame sequences, producing two groups of responses. A positive reward is assigned only when the proportion of correct answers from the ordered group exceeds that from the shuffled one. This strategy encourages the model to exploit temporal reasoning policy rather than relying on shortcuts derived from isolated frames.

Besides, to tackle the scarcity of high-quality video reasoning data, we strategically introduce image-based reasoning data as part of training data. We construct two datasets: Video-R1-CoT-165k for SFT cold start and Video-R1-260k for RL training. The image data serves as a valuable foundation for training general reasoning skills, while the curated video samples provide the temporal complexity needed for video understanding. This hybrid training setup not only alleviates the data bottleneck but also enables the model to transfer reasoning skills learned from static images to dynamic video contexts. Combined with T-GRPO, this approach equips Video-R1 with stronger, more generalizable video reasoning capabilities.

Our experiments show that Video-R1 achieves consistent and significant improvements across a suite of challenging video reasoning benchmarks, including VSI-Bench [38], VideoMMMU [13], MMVU [48], MVBench [20], TempCompass [27], and VideoMME [9]. Notably, Video-R1-7B attains 37.1% accuracy on VSI-Bench, a challenging video spatial reasoning benchmark, outperforming even proprietary models like GPT-4o [15]. These results suggest that with carefully designed algorithms and data pipelines, RL can indeed unlock complex temporal reasoning capabilities in MLLMs, similar to the breakthroughs seen in the text domain. Our contributions can be summaried as follows:

- We propose **Video-R1**, as the first attempt to systematically to explore developing video reasoning MLLMs based on the R1 paradigm. To support training, we construct two reasoning datasets: **Video-R1-CoT-165k** for SFT and **Video-R1-260k** for RL training, incorporating both image and video reasoning samples. We hope that Video-R1 will serve as a foundation for future research on video reasoning.

- To address the lack of temporal modeling in existing RL methods, we introduce **T-GRPO**, a novel training algorithm that encourages the model to utilize temporal information by contrasting reasoning performance over ordered and shuffled video frames.

- Extensive experiments on multiple video benchmarks, such as VideoMMMU, VSI-Bench, MVBench, etc, demonstrate the effectiveness of our approach. Notably, **Video-R1-7B** achieves **37.1%** accuracy on VSI-Bench, outperforming the proprietary GPT-4o model.

## 2 Related Works

### 2.1 Multimodal Large Language Models for Video

Video understanding is an essential capability for Multimodal Large Language Models (MLLMs), enabling them to interpret and reason over dynamic visual content [31, 4, 37, 44, 46, 32, 2]. In recent years, a number of MLLMs have been developed specifically to advance progress in video understanding tasks. For example, LLaMA-VID [22] proposes a dual-token strategy (context and content tokens) to compress video input representations, enabling vision-language models to efficiently handle long videos while retaining essential visual information. VideoLLaMA2 [4] enhances video-language modeling by introducing spatial-temporal convolution for better dynamic understanding and an audio branch to integrate multimodal cues for richer video comprehension. LongVA [43] extends the context window of language backbones to process significantly longer video sequences without specialized video training, offering a language-centric solution to long-range temporal reasoning. VISA [37] introduces a knowledge-driven video object segmentation task that combines world knowledge with object tracking, addressing implicit, complex video queries through a segmentation-enabled multimodal LLM. These advancements highlight the potential of MLLMs in advancing video understanding. However, most prior works have primarily focused on video perception tasks. The development of MLLMs with strong video reasoning capabilities remains largely unexplored.

### 2.2 Large Language Model Reasoning

The reasoning abilities of Large Language Models (LLMs) have been a focal point of recent research, aiming to enhance their capacity to perform complex, multi-step problem-solving tasks [34, 47, 42, 49, 23, 8, 35, 41, 24]. Unlike earlier approaches that rely on dense, step-level supervision or learned reward models to supervise reasoning paths [10, 21], DeepSeek-R1 initiates a new wave of interest in rule-based reinforcement learning, demonstrating that even coarse, outcome-only rewards can effectively elicit strong reasoning behavior [11]. Its success shows that with a carefully designed reward structure and policy optimization strategy, models could learn to generate long CoT without requiring intermediate supervision. Following this paradigm, several recent efforts have attempted to reproduce R1's success [14, 33, 36, 3, 28, 18, 6, 45]. For example, Open Reasoner Zero [12] and Kimi k1.5 [33] explore similar rule-based RL pipelines to enhance reasoning in the text and image domains, respectively. However, despite encouraging progress, few prior work has explored how to extend this approach to the video domain. Bridging this gap remains an open challenge and a promising direction for expanding the boundaries of reasoning models.

## 3 Methods

### 3.1 Dataset Construction

High-quality training data plays a crucial role in reinforcing video reasoning capabilities in MLLMs. In this section, we will introduce how we curate Video-R1-260k for RL training and Video-R1-CoT-165k for SFT cold start.

**Data Collection and Curation.**   To overcome the scarcity of high-quality video reasoning training data, we strategically introduce image-based reasoning data as part of training data. The image-based data serves primarily to teach the model a broad range of reasoning skills, covering various difficulty levels and domains such as math, spatial logic, expert-level knowledge, etc. These samples help the model develop generalized reasoning abilities in static contexts. In contrast, the video-based data is primarily used to train the model's ability to perform temporal reasoning—including understanding event progression, capturing frame-to-frame dependencies, and drawing inferences based on motion and causal dynamics over time.

We collect data from a variety of public datasets and carefully sample and balance the proportion of each subset. The final composition of the Video-R1-260k dataset is illustrated in Figure 2. The distribution of Video-R1-260k dataset can be roughly categorized as follows:

- **General (Video, 116k):** A diverse set of open-domain video data, covering everyday scenarios and designed to build temporal comprehension and reasoning abilities.

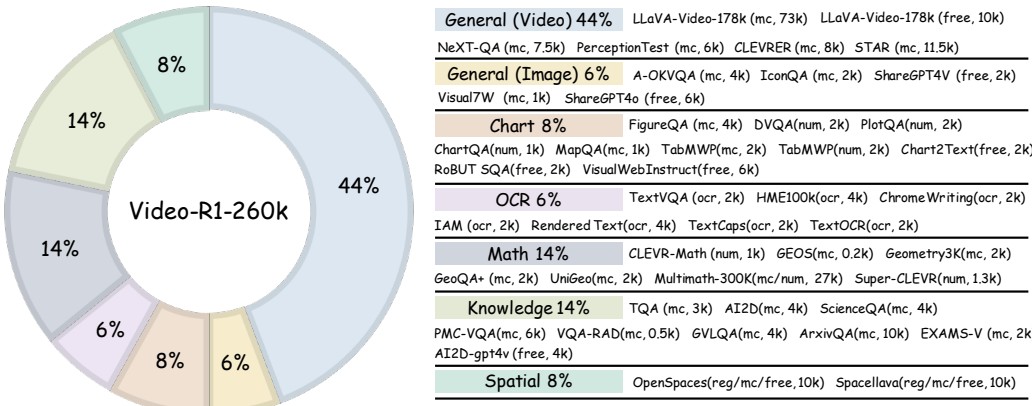

Figure 2: The data distribution of our Video-R1-260k dataset.

- **General (Image, 15k):** A general-purpose image question-answering data, used to provide basic visual understanding.
- **Chart (Image, 21k):** Visual reasoning over charts, line graphs and scientific figures, focusing on data interpretation and quantitative logic.
- **OCR (Image, 16k):** Facilitate reasoning tasks that require recognizing and interpreting embedded textual content such as signs, forms, or documents.
- **Math (Image, 37k):** Image-based math reasoning questions, including formulas, geometry diagrams, and multi-step symbolic reasoning.
- **Knowledge (Image, 37k):** Visual commonsense and multi-discipline reasoning tasks, testing the model's ability to integrate world knowledge with visual cues.
- **Spatial (Image, 20k):** Tasks that require understand spatial information for reasoning.

**CoT Annotation.**    To facilitate an effective SFT cold start, we leverage Qwen2.5-VL-72B-Instruct [1] to generate CoT rationales for the samples in Video-R1-260k. The prompt template for CoT generation is provided in Appendix C.1. After applying basic rule-based filtering to remove low-quality or inconsistent outputs, we obtain a high-quality CoT dataset, Video-R1-CoT-165k, which is used for the cold-start SFT stage.

**Data Type and Rule-based Reward Design.**    Since our reinforcement learning framework follows the rule-based reward paradigm of DeepSeek-R1 [11], it is crucial to ensure that the reward signals are both reliable and precise. To this end, the majority of our training data is designed around tasks with clearly verifiable outputs, such as multiple-choice and numerical answer formats. This allows for accurate reward computation using simple rules, ensuring stable and effective RL training.

However, to increase the model's flexibility and its ability to generalize across diverse tasks and formats, we also incorporate a smaller portion of other data types. These include free-form generation, OCR tasks, and regression problems, which are essential for adapting to real-world applications and broader datasets.

The data types and corresponding correctness reward functions are summarized as follows:

- **Multiple Choice:** The reward is assigned based on whether the predicted answer matches the ground-truth answer.
- **Numerical QA:** A binary reward is given depending on whether the predicted number exactly matches the reference value.
- **OCR:** We compute the reward using the Word Error Rate (WER), measuring the edit distance between the predicted and reference text.
- **Free-form QA:** The reward is calculated as the average of ROUGE-1, ROUGE-2, and ROUGE-L scores between the model's output and the ground-truth answer.
- **Regression:** The closer the predicted value is to the ground truth, the higher the reward, calculated as one minus their relative error.

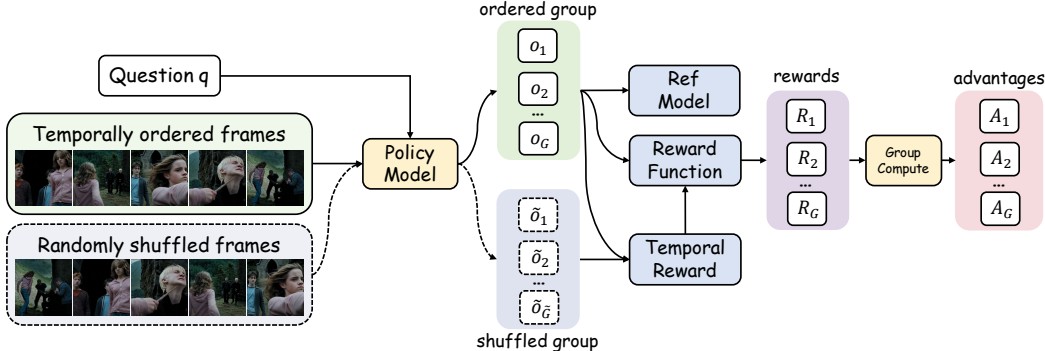

Figure 3: An illustration of our proposed T-GRPO algorithm.

## 3.2 Temporal Group Relative Policy Optimization (T-GRPO)

While GRPO [30] has proven effective in text-based reasoning, it lacks explicit reward signals for temporal reasoning—making it insufficient for training MLLMs to reason over videos. To address this, we propose Temporal Group Relative Policy Optimization (T-GRPO), which introduces a contrastive reward mechanism that explicitly encourages temporal reasoning, as illustrated in Figure 3.

The core idea behind T-GRPO is to compare the model's performance on the same video question when frames are provided in two different orders: (1) the *temporally ordered* sequence, and (2) a *randomly shuffled* version. For each input question, we generate two groups of responses $\{o_i\}_{i=1}^{G}$ and $\{\tilde{o}_i\}_{i=1}^{\tilde{G}}$ using the ordered and shuffled frame inputs, respectively.

Let $p$ and $\tilde{p}$ denote the proportion of correct answers in each group. We then define a temporal reward $r_t$ as:

$$r_t = \begin{cases} \alpha, & \text{if } p \geq \tilde{p} \\ 0, & \text{otherwise} \end{cases} \tag{1}$$

where $\alpha$ is a hyperparameter controlling the magnitude of the temporal reward. Here we set $\alpha = 0.3$.

This contrastive design encourages the model to perform better when the video is presented in correct temporal order than when it is shuffled. The model is only granted this positive reward if its current reasoning strategy for a given question demonstrates a reliance on temporal information. For tasks with continuous rewards (e.g., free-form answers), a threshold (e.g., 0.5) can be used to determine whether a response is considered correct.

Importantly, $r_t$ is only applied to correct responses to ensure meaningful positive advantages. Applying it to all responses would dilute the reward signal and hinder effective learning. In other words, when the model's reasoning policy successfully relies on temporal patterns, correct responses are reinforced with a higher reward, while incorrect ones remain unaffected.

Formally, the temporal-augmented reward is defined as:

$$R_i = \begin{cases} r_i + r_t, & \text{if } o_i \text{ is correct} \\ r_i, & \text{otherwise} \end{cases} \tag{2}$$

where $r_i$ is the reward for response $i$, containing both the correctness reward and the format reward, following [11]. $R_i$ is the final reward used for calculating advantages. This reward shaping ensures that when the model answers correctly under a temporal setting but fails to outperform the shuffled baseline, it receives no additional reward—pushing the optimization toward adopting a more temporally aware reasoning policy. The temporal reward $r_t$ could also be added to the advantages directly.

Then, the advantage $A_i$ is computed over the rewards within each group:

$$A_i = \frac{R_i - \text{mean}(\{R_j\})}{\text{std}(\{R_j\})} \tag{3}$$

Following DeepSeek R1 [11], the final policy update is as follows:

$$\mathcal{J}_{\text{T-GRPO}}(\theta) = \mathbb{E}_{q,\{o_i\}} \left[ \frac{1}{G} \sum_{i=1}^{G} \Big( \min \Big( \frac{\pi_\theta(o_i|q)}{\pi_{\theta_{\text{old}}}(o_i|q)} A_i, \quad \text{clip} \Big( \frac{\pi_\theta(o_i|q)}{\pi_{\theta_{\text{old}}}(o_i|q)}, 1 - \epsilon, 1 + \epsilon \Big) A_i \Big) \right.$$
$$\left. - \beta \, \mathbb{D}_{\text{KL}}(\pi_\theta \| \pi_{\text{ref}}) \Big) \right] \tag{4}$$

By explicitly comparing the model's performance under ordered and shuffled inputs, T-GRPO introduces a contrastive training signal that drives the model to prefer reasoning strategies that leverage temporal patterns. It is worth noting that T-GRPO is only employed for video-based inputs in the training process of Video-R1.

### 3.3 Training Strategies

We adopt Qwen2.5-VL-7B-Instruct [1] as the base MLLMs for training. Similar to DeepSeek-R1 [11], the training process is conducted in two stages: SFT cold start followed by RL training. For these two stages, we both adopt image-video mixed training strategy.

In the first stage, we perform SFT on the Video-R1-CoT-165k dataset. This step serves as a cold-start initialization, equipping the model with basic reasoning capabilities across a variety of modalities. The resulting model is denoted as Qwen2.5-VL-7B-SFT.

In the second stage, we further train the Qwen2.5-VL-7B-SFT model on the broader Video-R1-260k dataset using our proposed T-GRPO algorithm. This reinforcement learning phase is designed to guide the model beyond the rigid, pattern-matching behavior induced by SFT, encouraging it to freely explore better reasoning strategies. The resulting model is denoted as Video-R1-7B.

To further enhance the quality of reasoning, we introduce a length-based reward to regulate the length of the model's output. Specifically, this mechanism aims to strike a balance between encouraging deeper reasoning and preventing overthinking. For each reasoning path $o_i$, if the predicted answer is correct and the response length falls within a predefined interval $[l_{\min}, l_{\max}]$, the model receives an additional reward $r_l = \omega$. Formally:

$$R_i = \begin{cases} R_i + \omega, & \text{if } o_i \text{ is correct and } l_{\min} \leq \text{len}(o_i) \leq l_{\max} \\ R_i, & \text{otherwise} \end{cases} \tag{5}$$

This reward encourages the model to think deeply without overthinking. In this paper, we set $\omega = 0.2$, $l_{min} = 320$ and $l_{max} = 512$. The analysis of this reward's effect is provided in Appendix A.2.

### 3.4 Aha Moment in Video Reasoning

We find that Video-R1 is capable of performing complex reasoning with self-reflective behaviors, often referred to as "aha moments" [11]. Specifically, the model occasionally revisits its interpretation of the video or reflects on previous outputs, particularly when confronted with ambiguous temporal cues or multi-step inference tasks. These behaviors indicate that the model is not merely executing memorized patterns but is actively engaging in internal feedback loops by re-examining evidence. An example is demonstrated in Figure 4. More examples can be found in Appendix B.

## 4 Experiments

### 4.1 Setup

**Benchmarks.** We evaluate our model on six video benchmarks: VSI-Bench [38], VideoMMMU [13], MMVU [48], MVBench [20], TempCompass [27], and VideoMME [9]. Among them, the first three are video reasoning benchmarks, which focus primarily on assessing the model's reasoning capabilities in video understanding. The latter three are general-purpose video understanding benchmarks, which include a mixture of perception and reasoning tasks. For MMVU, we evaluate on its multiple-choice question set for stability and consistency. For all evaluations, we follow the decoding configuration used in the official Qwen2.5-VL demo, with top_p = 0.001 and temperature = 0.01.

Figure 4: An example of Video-R1-7B's reasoning output on VSI-Bench.

**Training Details.** We train our model using up to 8 NVIDIA A100 (80GB) GPUs. For efficiency considerations, we limit the maximum number of video frames to 16 during training. Each frame is processed at a max resolution of $128 \times 28 \times 28$ pixels. During inference, we increase the frame resolution to $256 \times 28 \times 28$ pixels and frames to 16~64 to enhance performance. The ordered group size $G$ is set to 8 and the shuffled group size $\tilde{G}$ is set to half of that for efficiency. More details are provided in Appendix D.

We first perform SFT on Video-R1-CoT-165k for one epoch to obtain the Qwen2.5-VL-7B-SFT model. This is followed by RL training on Video-R1-260k to produce the final Video-R1 model. Due to current computational resource limitations, we train the model for only 1k RL steps. Surprisingly, even within this limited training budget, the model exhibits significant improvements in video reasoning performance, indicating the strong effectiveness of both our data design and algorithm. Further experiments exploring the impact of scaling up RL training are presented in Appendix A.1.

## 4.2 Main Results

As shown in Table 1. Our experimental results across six benchmarks validate the effectiveness of Video-R1 in video reasoning and general video understanding tasks. The key findings are as follows.

**Superior Performance of Video-R1.** Video-R1 significantly outperforms previous models across most benchmarks, with particularly strong gains on video reasoning tasks such as VSI-Bench, VideoMMMU, and MMVU. This highlights the necessity of explicit reasoning capability in solving video tasks, and confirms the effectiveness of reinforcement learning for video tasks.

**RL Works Better Than SFT.** We observe that the SFT model Qwen2.5-VL-7B-SFT does not consistently improve performance. In some cases (e.g., VideoMME), performance even slightly drops after SFT, likely due to overfitting or limited generalization in unseen scenarios [5]. In contrast, after only 1k steps of reinforcement learning, Video-R1 achieves significant performance boosts. This clearly demonstrates the importance of RL in unlocking generalizable video reasoning capability.

Table 1: Performance of different models on benchmarks.

| Models | Frames | Video Reasoning Benchmark | | | Video General Benchmark | | |
| --- | --- | --- | --- | --- | --- | --- | --- |
| | | VSI-Bench | VideoMMMU | MMVU (mc) | MVBench | TempCompass | VideoMME (wo sub) |
| GPT-4o [15] | - | 34.0 | 61.2 | 75.4 | - | - | 71.9 |
| LLaMA-VID [22] | - | - | - | - | 41.9 | 45.6 | - |
| VideoLLaMA2 [4] | - | - | - | 44.8 | 54.6 | - | 47.9 |
| LongVA-7B [43] | - | 29.2 | 23.9 | - | - | 56.9 | 52.6 |
| VILA-1.5-8B [25] | - | 28.9 | 20.8 | - | - | 58.8 | - |
| VILA-1.5-40B [25] | - | 31.2 | 34.0 | - | - | - | 60.1 |
| Video-UTR-7B [40] | - | - | - | - | 58.8 | 59.7 | 52.6 |
| LLaVA-OneVision-7B [19] | - | 32.4 | 33.8 | 49.2 | 56.7 | - | 58.2 |
| Kangaroo-8B [26] | - | - | - | - | 61.1 | 62.5 | 56.0 |
| Qwen2.5-VL-7B (CoT) | 16 | 27.7 | 47.8 | 59.2 | 57.4 | 72.2 | 53.1 |
| Qwen2.5-VL-7B-SFT | 16 | 31.8 | 47.4 | 61.3 | 59.4 | 69.2 | 52.8 |
| Qwen2.5-VL-7B (CoT) | 32 | 30.1 | 48.1 | 60.0 | 59.0 | 72.6 | 56.6 |
| Qwen2.5-VL-7B-SFT | 32 | 33.3 | 49.4 | 63.5 | 60.5 | 69.9 | 55.4 |
| Qwen2.5-VL-7B (CoT) | 64 | 31.4 | 50.4 | 60.0 | 59.2 | 72.9 | 59.6 |
| Qwen2.5-VL-7B-SFT | 64 | 34.8 | 49.4 | 61.6 | 60.6 | 70.0 | 58.8 |
| Video-R1-7B | 16 | 34.6 | 49.8 | **64.2** | 62.7 | 72.6 | 57.4 |
| Video-R1-7B | 32 | 35.8 | 52.3 | 63.8 | 63.9 | 73.2 | 59.3 |
| Video-R1-7B | 64 | **37.1** | **52.4** | 63.8 | **64.8** | **73.2** | **61.4** |

Table 2: Ablation Study.

| Models | Frames | Video Reasoning Benchmark | | | Video General Benchmark | | |
| --- | --- | --- | --- | --- | --- | --- | --- |
| | | VSI-Bench | VideoMMMU | MMVU (mc) | MVBench | TempCompass | VideoMME (wo sub) |
| Video-R1-7B-wo-image | 16 | 32.3 | 45.8 | 60.6 | 60.9 | 69.8 | 53.8 |
| Video-R1-7B-wo-temporal | 16 | 32.7 | 48.3 | 62.1 | 61.1 | 71.3 | 54.5 |
| Video-R1-7B-zero | 16 | 31.8 | 49.5 | 63.8 | 60.4 | 70.9 | 53.8 |
| Video-R1-7B | 16 | 34.6 | 49.8 | 64.2 | 62.7 | 72.6 | 57.4 |

**More Frames Lead to Better Reasoning.** When increasing the input frame number, we observe performance improvements on almost all benchmarks. This indicates that richer context and temporal information contribute positively to the reasoning performance. Therefore, developing models capable of understanding and reasoning over longer video inputs is a promising direction for future research.

### 4.3 Ablation Study

In this section, we conduct an ablation study by designing three variants of our model: Video-R1-7B-wo-image, which removes all image-based data during training and relies solely on video data; Video-R1-7B-wo-temporal, which replaces our proposed T-GRPO algorithm with the original GRPO method; Video-R1-zero, which skips the SFT cold start and directly conducts RL training.

As shown in Table 2, both ablated models perform worse than the full Video-R1-7B across all benchmarks. In particular, removing image data leads to a noticeable drop in performance on both video reasoning and general benchmarks, indicating that image-based samples play a crucial role in bootstrapping general reasoning ability. Similarly, without temporal-aware training via T-GRPO, the model struggles to fully leverage temporal cues, resulting in weaker performance on benchmarks. Additionally, skipping the SFT cold start leads to worse performance, underscoring the value of our Video-R1-CoT-165k dataset and the importance of initializing with SFT before RL training. These ablations validate the effectiveness of our proposed methods.

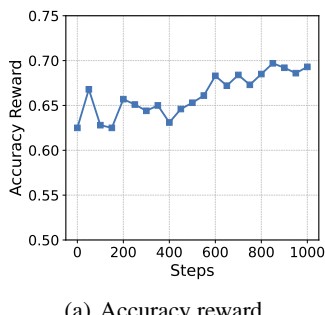
(a) Accuracy reward

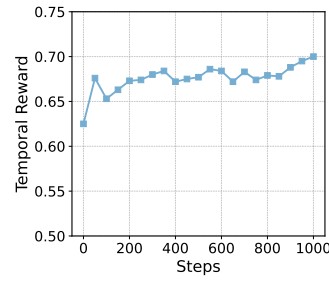
(b) Temporal reward $r_t$ (scaled)

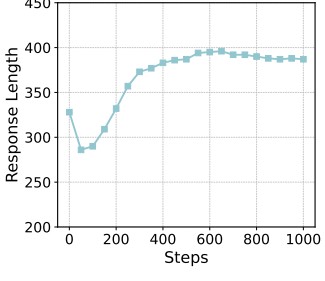
(c) Response length

Figure 5: RL training curves.

## 4.4 Training Curves

Figure 5 illustrates the RL training dynamics of Video-R1. As shown in Figure 5(a), the accuracy reward exhibits a generally upward trend, indicating that the model continuously improves its ability to produce correct answers under reinforcement learning. In Figure 5(b), the temporal reward $r_t$ (scaled to 0~1 for visibility) also demonstrates a steady increase. This suggests that the model is progressively adopting more temporally grounded reasoning strategies with T-GRPO. Interestingly, the response length in Figure 5(c) drops initially during RL training, then rises and stabilizes. We guess that this may reflect a learning transition: the model first discards its sub-optimal SFT reasoning style, and eventually settles on a new reasoning policy.

## 4.5 Effect of Temporal Reward Analysis

To further assess the impact of T-GRPO, we measure the percentage of responses that incorporate temporal reasoning for questions requiring it. Specifically, we use Qwen2.5-VL-72B to identify all temporally grounded questions across six benchmarks, and then use Qwen2.5-VL-72B to evaluate whether the model responses demonstrate temporal reasoning. The evaluation prompt can be

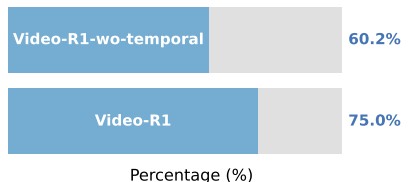

Figure 6: Percentage of temporal reasoning responses.

found in Appendix C.2. As shown in Figure 6, Video-R1 trained by T-GRPO incorporates temporal reasoning in 75.0% of responses, compared to 60.2% for Video-R1-wo-temporal, which lacks temporal modeling. This clear gap demonstrates the effectiveness of T-GRPO in encouraging the model to leverage temporal information during reasoning.

We also conduct a sensitivity analysis on the magnitude of the temporal reward, controlled by the hyperparameter $\alpha$, as shown in Figure 7. The average performance is calculated as the mean performance across six benchmarks. We observe a slight drop in performance at $\alpha = 0.1$ and $\alpha = 0.4$, while $\alpha = 0.2$ and $\alpha = 0.3$ yield similar and favorable results. This indicates that the model is relatively insensitive to the choice of $\alpha$ within a reasonable range.

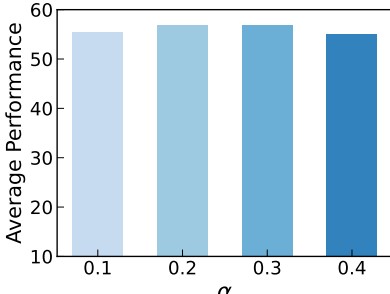

Figure 7: Sensitivity analysis.

## 5 Conclusions

In this work, we present Video-R1, as an attempt to investigate the R1 paradigm to enhances video reasoning in multimodal large language models. Motivated by DeepSeek-R1, we extend rule-based reinforcement learning to video by introducing T-GRPO, a temporal-aware extension of GRPO that explicitly encourages temporal reasoning in video. To support training, we curate two datasets, Video-R1-CoT-165k for SFT cold start and Video-R1-260k for RL. Experimental results across six benchmarks validate the effectiveness of our approach. We hope this work provides a foundation for further research in video reasoning with MLLMs.

# 6 Acknowledgment

This work is partially supported by the National Natural Science Foundation of China (Grant No. 62306261), and The Shun Hing Institute of Advanced Engineering (SHIAE) Grant (No. 8115074). This study was supported in part by the Centre for Perceptual and Interactive Intelligence, a CUHK-led InnoCentre under the InnoHK initiative of the Innovation and Technology Commission of the Hong Kong Special Administrative Region Government. This work is also partially supported by Hong Kong RGC Strategic Topics Grant STG1/E-403/24-N, and CUHK-CUHK(SZ)-GDST Joint Collaboration Fund YSP26-4760949.

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

# A  Additional Experiments

## A.1  Scaling Up RL Training

In previous experiments, our Video-R1 is trained with 1k RL steps, similar to prior works that apply the R1 paradigm in other domains [7, 29]. To further explore the impact of scaling up reinforcement learning, we extend the training to 10k RL steps. As shown in Table 3, the performance of Video-R1 generally improves on various benchmarks with more RL training steps. These results demonstrate the effectiveness of large-scale RL training and suggest that additional training can further enhance the model's reasoning capabilities.

Table 3: Results on more RL training steps.

| Models | Frames | Video Reasoning Benchmark | | | Video General Benchmark | | |
| --- | --- | --- | --- | --- | --- | --- | --- |
| | | VSI-Bench | VideoMMMU | MMVU (mc) | MVBench | TempCompass | VideoMME (wo sub) |
| Video-R1-7B (1k steps) | 16 | 34.6 | 49.8 | 64.2 | 62.7 | 72.6 | 57.4 |
| Video-R1-7B (1k steps) | 32 | 35.8 | 52.3 | 63.8 | 63.9 | 73.2 | 59.3 |
| Video-R1-7B (1k steps) | 64 | 37.1 | 52.4 | 63.8 | 64.8 | 73.2 | 61.4 |
| Video-R1-7B (10k steps) | 16 | 34.5 | 50.7 | 64.2 | 63.7 | 73.7 | 57.2 |
| Video-R1-7B (10k steps) | 32 | 35.7 | **52.7** | **66.2** | 64.7 | 74.2 | 59.3 |
| Video-R1-7B (10k steps) | 64 | **37.8** | 51.4 | 65.0 | **65.5** | **74.2** | **61.8** |

## A.2  Effect of Length Reward Analysis

To evaluate the impact of the proposed length reward, we conduct an ablation study by comparing Video-R1 with a variant trained without the length reward, denoted as Video-R1-wo-len. As shown in Figure 8(b), removing the length reward leads to a clear decline in response length during RL training, whereas Video-R1 maintains a steadily increasing length that eventually stabilizes at a higher level. Finally, Video-R1-wo-len achieves lower average performance across benchmarks compared to Video-R1, as shown in Figure 8(a). We guess that this is because the model fails to preserve a deep reasoning style without the length reward during training, instead favoring shorter and potentially less informative responses. This likely results in weaker generalization during evaluation. These findings highlight the importance of encouraging the model to maintain a moderate level of reasoning effort during training.

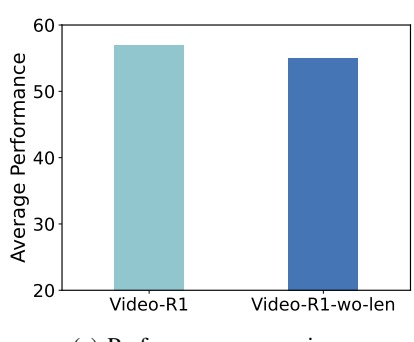

(a) Performance comparison

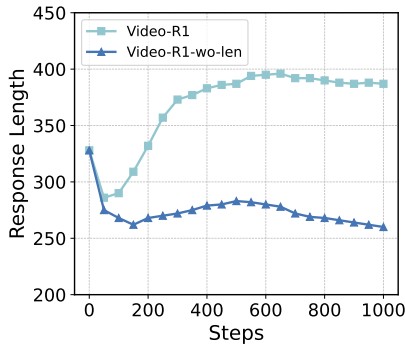

(b) Length curve comparison

Figure 8: Effect of length reward.

## B Reasoning Examples

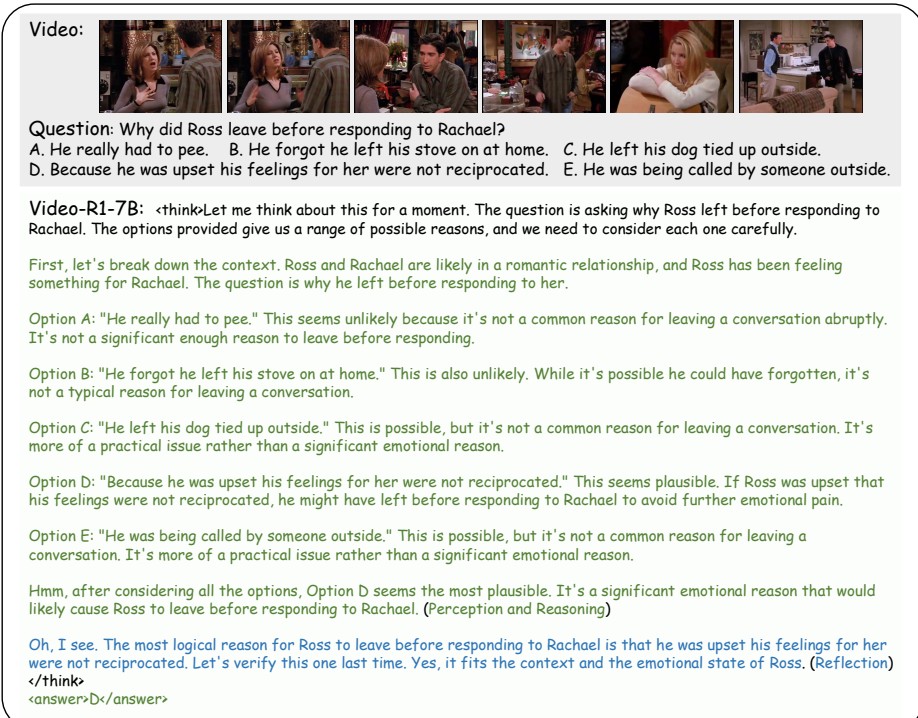

Video:

Question: Which move motion in the video lose the system energy?

Video-R1-7B: <think>Let me think about this for a moment. The question is asking about which move in the video loses kinetic energy. To answer this, I need to carefully analyze the video and identify the different actions and their outcomes.

First, let's break down the video content. The video shows a series of actions involving a pool table, a bowling alley, and a car crash. In the pool table scene, a player takes a shot, and the balls move around. In the bowling alley scene, a ball rolls down the lane and hits the pins. In the car crash scene, two cars collide.

Now, let's consider the energy loss in each scene:
1. In the pool table scene, the player takes a shot, and the balls move. The kinetic energy is transferred from the cue stick to the balls, and the balls continue to move. This does not seem to lose kinetic energy.
2. In the bowling alley scene, the ball rolls down the lane and hits the pins. The ball loses some kinetic energy upon hitting the pins, but it still has some energy left as it rolls away.
3. In the car crash scene, the two cars collide. The kinetic energy is transferred between the cars, and some energy is lost as heat and deformation of the cars. This is a significant loss of kinetic energy.
Hmm, it seems that the car crash scene is the one where the most kinetic energy is lost. The cars collide, and the energy is dissipated in the form of heat and deformation, which is a significant loss compared to the other scenes. (Perception and Reasoning)

Oh, I see. The car crash scene is the one where the most kinetic energy is lost. The other scenes involve transfers of energy, but the car crash involves a significant loss.

Let's verify this. The car crash scene is indeed the one where the most kinetic energy is lost, as the cars collide and the energy is dissipated in a dramatic way. (Reflection)

So, the move that loses the most kinetic energy in the video is the car crash.</think>

<answer>The car crash in the video loses the system energy.</answer>

Figure 9: An example of Video-R1-7B's reasoning output on MMVU Benchmark.

Video:

Question: Why did Ross leave before responding to Rachael?
A. He really had to pee.    B. He forgot he left his stove on at home.    C. He left his dog tied up outside.
D. Because he was upset his feelings for her were not reciprocated.    E. He was being called by someone outside.

Video-R1-7B: <think>Let me think about this for a moment. The question is asking why Ross left before responding to Rachael. The options provided give us a range of possible reasons, and we need to consider each one carefully.

First, let's break down the context. Ross and Rachael are likely in a romantic relationship, and Ross has been feeling something for Rachael. The question is why he left before responding to her.

Option A: "He really had to pee." This seems unlikely because it's not a common reason for leaving a conversation abruptly. It's not a significant enough reason to leave before responding.

Option B: "He forgot he left his stove on at home." This is also unlikely. While it's possible he could have forgotten, it's not a typical reason for leaving a conversation.

Option C: "He left his dog tied up outside." This is possible, but it's not a common reason for leaving a conversation. It's more of a practical issue rather than a significant emotional reason.

Option D: "Because he was upset his feelings for her were not reciprocated." This seems plausible. If Ross was upset that his feelings were not reciprocated, he might have left before responding to Rachael to avoid further emotional pain.

Option E: "He was being called by someone outside." This is possible, but it's not a common reason for leaving a conversation. It's more of a practical issue rather than a significant emotional reason.

Hmm, after considering all the options, Option D seems the most plausible. It's a significant emotional reason that would likely cause Ross to leave before responding to Rachael. (Perception and Reasoning)

Oh, I see. The most logical reason for Ross to leave before responding to Rachael is that he was upset his feelings for her were not reciprocated. Let's verify this one last time. Yes, it fits the context and the emotional state of Ross. (Reflection)
</think>
<answer>D</answer>

Figure 10: An example of Video-R1-7B's reasoning output on MVBench.

## C Prompt Template

### C.1 Prompt Template for Training and Inference

Figure 11 illustrates the prompt template for training and inference of all models. We also use this prompt for the COT annotation.



**Prompt Template for Training and Inference**

{Question}
Please think about this question as if you were a human pondering deeply. Engage in an internal dialogue using expressions such as 'let me think', 'wait', 'Hmm', 'oh, I see', 'let's break it down', etc, or other natural language thought expressions. It's encouraged to include self-reflection or verification in the reasoning process. Provide your detailed reasoning between the <think> and </think> tags, and then give your final answer between the <answer> and </answer> tags.
{OutputTemplate}

OutputTemplate:
"multiple choice": Please provide only the single option letter (e.g., A, B, C, D, etc.) within the <answer> </answer> tags.
"numerical": Please provide the numerical value (e.g., 42 or 3.14) within the <answer> </answer> tags.
"OCR": Please transcribe text from the image/video clearly and provide your text answer within the <answer> </answer> tags.
"free-form": Please provide your text answer within the <answer> </answer> tags.
"regression": " Please provide the numerical value (e.g., 42 or 3.14) within the <answer> </answer> tags.



Figure 11: Prompt template for training and inference.

### C.2 Prompt Template for Temporal Reasoning Evaluation

Figure 12 illustrates the prompt template for temporal reasoning evaluation.



**Prompt Template for Temporal Reasoning Evaluation**

{Question}
You are given a question about a video and a response to that question. Your task is to assess whether the response engages with any aspect of the video's temporal structure — that is, any elements related to time, sequence, or progression. This may include references to actions occurring in a particular order, events unfolding over time, descriptions of what happens before or after, mentions of duration, speed, or direction, or any general sense of temporal flow conveyed in the response. Consider a broad range of interpretations, including responses that reflect an understanding of how events evolve or follow one another. Provide your reasoning between the <think> and </think> tags, and then write either 'yes' or 'no' between the <answer> and </answer> tags.



Figure 12: Prompt template for temporal reasoning evaluation.

## D Additional Implementation Details

We use the Adam optimizer with a learning rate of 1e-6 to train our model. The SFT stage takes approximately 40 hours per epoch, while the RL stage takes around 15 hours for 1k steps. The hyperparameter $\beta$ in the KL divergence term of the GRPO algorithm is set to 0.04. To ensure training stability, we apply a weight decay rate of 0.01 and clip the maximum gradient norm to 5. The maximum response length is set to 768 tokens.

# E  Limitations and Future Works

We envision this work as a foundation for advancing research in video reasoning with MLLMs. Below, we outline its limitations and potential avenues for future work:

- **Increasing Frames Number.** Currently, our model is trained with 16 video frames, which may limit its ability to handle long-range temporal dependencies. In future work, we can develop more efficient training and inference strategies that allow scaling to longer videos, enabling more comprehensive temporal reasoning.

- **Better Temporal Modeling Method.** Although T-GRPO introduces effective temporal-aware reasoning, it brings additional computational overhead due to contrastive evaluation and reward calculation. This could be mitigated through inference acceleration framework such as vLLM, or by exploring more efficient mechanisms for temporal modeling.

- **Adaptive Response Length Control.** Our current length control mechanism applies a fixed reward within a predefined range, regardless of the complexity of each sample. Future work could explore dynamic length control strategies, where the model adaptively determines the appropriate response length based on the difficulty or type of the question.

- **Refined Image-to-Video Knowledge Transfer.** At present, we incorporate image-based reasoning data in a straightforward manner by mixing it into the training set. Future research could design more principled approaches for leveraging image data to more effectively transfer reasoning ability from images to videos.

- **Generalist Video Reward Modeling.** Currently, we use rule-based reward functions tailored to different tasks. A promising future direction is to develop a generalist video reward model, which is capable of providing consistent and scalable reward signals across various video reasoning tasks. Such a model could reduce reliance on handcrafted rules and enable broader applicability of reinforcement learning in video understanding.

