# OpenReview forum: "Video-R1: Reinforcing Video Reasoning in MLLMs"
_NeurIPS.cc/2025/Conference — NeurIPS 2025 poster_

### Official Review · Reviewer_jHCR · 2025-06-21

**Clarity:** 2
**Significance:** 4
**Originality:** 3
**Rating:** 5
**Confidence:** 4

**Summary:**

The paper introduces an R1-style framework for video reasoning task.
The main contributions, as mentioned in the paper, are as follows:
1. Introduce T-GRPO, an adaptation of GRPO for temporal-based video reasoning.
2. Video-R1, a Qwen-based model trained using T-GRPO for video reasoning.
3.  2 datasets for cold starting and RL training on video reasoning tasks.

The paper provides detailed information about the T-GRPO algorithm. It uses the GRPO algorithm allong with an additional reward based on reasoning using the right order of frames. This forces the model to use temporal information during reasoning allowing for accurate outputs.

The contribution of the datasets allows for exapnding this methodology with outher foundation models as the backbone as well as addding to the pool of present data for future works.

The authors of the work undertake extensive experimentation on multiple video reasoning datasets.
The work shows a significant performance increase on various video reasoning datasets using their methodology, showing the viability of their method with a strong ablation study measuring the upliftment due to each part of their framework.

**Questions:**

The same as described in the weaknesses.

**Ethical Concerns:**

["NO or VERY MINOR ethics concerns only"]

**Final Justification:**

The authors have addressed my concerns and believe that the work shines light on a promising direction to push the field of video reasoning.

**Limitations:**

yes

**Quality:**

4

**Strengths And Weaknesses:**

Strengths:
1.  The paper introduces a new framework and potential direction for video reasoning.
2. The paper shows significant performance in various datasets.
3. The work presents extensive experimentation and ablation study along with a detailed appendix.

Weakness:
1. Some variables in the equations are not clearly described or assume prior knowledge, making it difficult to follow. For example,
    a. In Equation 2, it is unclear whether $R_i$ is calculated as the reward for each output or the overall set of outputs. \
    b. In Equation 3, $\pi_{ref}$ is not defined when describing the GRPO algorithm.

While the definition of variables in other places can be inferred from context, it leaves room for ambiguity, leading to difficulty in reproducing the results.

---

> ### Author Rebuttal · Authors · 2025-07-30
>
> > Q1：Some variables in the equations are not clearly described or assume prior knowledge, making it difficult to follow. For example, a. In Equation 2, it is unclear whether  is calculated as the reward for each output or the overall set of outputs. b. In Equation 3,  $pi_{ref}$ is not defined when describing the GRPO algorithm.
> >
> > While the definition of variables in other places can be inferred from context, it leaves room for ambiguity, leading to difficulty in reproducing the results.
>
> A1: Sorry for confusing you. Regarding Equation (2), the reward $r_i$ is computed for each output $o_i$, not for the question $q_i$ as incorrectly stated in the current version. We will correct this typo and revise the explanation for clarity. For Equation (3), following DeepSeek-R1, $\pi_{ref}$ refers to the policy of reference model used to compute the KL divergence. This term helps stabilize RL training by preventing the learned policy from drifting too far from the initial model. We will carefully revise the manuscript to ensure that all variables are explicitly defined and clearly explained.

---

> > ### Comment · Reviewer_jHCR · 2025-08-05
> >
> > Thank you for the prompt reply and clarification.
> >
> > I have no other issues with the work

---

> > > ### Author Response · Authors · 2025-08-05
> > >
> > > Thanks for your feedback. It’s great to hear that your concerns have been successfully addressed.

---

### Official Review · Reviewer_oq4J · 2025-06-30

**Clarity:** 3
**Significance:** 3
**Originality:** 2
**Rating:** 4
**Confidence:** 5

**Summary:**

In this work, the authors propose a Video-R1 paradigm for reinforcing video reasoning with MLLMs, inspired by the learning style of DeepSeek-R1. First, they build two reasoning datasets, i.e., Video-R1-CoT-165k and Video-R1-260k, respectively for SFT and RL training. Second, they introduce a T-GRPO learning method that leverages temporal modeling in RL to enhance video reasoning.

**Questions:**

Please see weakness section.

**Ethical Concerns:**

["NO or VERY MINOR ethics concerns only"]

**Final Justification:**

Thanks for the rebuttal. The authors have addressed most of my main concerns. I keep my original rating.

**Limitations:**

Please see weakness section.

**Quality:**

3

**Strengths And Weaknesses:**

*Strengths

1 The main contribution of this work is an early attempt on video reasoning. Different from the previous works, this paper systematically explores the R1 learning paradigm on video MLLMs,  based on SFT (with Video-R1-CoT-165k) and RL training (with Video-R1-260k). It would benefit video understanding community from basic perception towards deep thinking on complex video content.

2 The claims of this work are basically supported by their experiments on six video benchmarks: VSI-Bench, VideoMMMU, MMVU, MVBench, TempCompass, and VideoMME. Especially on video reasoning benchmarks like VSI-Bench, VideoMMMU and MMVU, the proposed Video-R1 achieves a preferable performance than recent open-source models and baselines, showing its effectiveness.

3 The structure of this paper is basically well organized.



*Weaknesses

1 The temporal reward is mainly based on frame order. The order-relevant supervision has been widely investigated in the video undestanding community such as [1-3].

2 The evalution is mainly based on video question answering. There are more temporal-relavent tasks such as temporal grounding and object tracking, which should be further investigated like VideoChat-R1 [4].

3 Small Concern: What is Qwen2.5-VL-7B (CoT)? Moreover, Qwen2.5-VL-7B (CoT) is generally worse than Qwen2.5-VL-7B-SFT on video reasoning benchmarks, while it is often better than Qwen2.5-VL-7B-SFT  on video general benchmarks such as TempCompass and VideoMME. Why this happens?

*Reference

[1] Shuffle and Learn: Unsupervised Learning using Temporal Order Verification, ECCV2016

[2] Unsupervised Representation Learning by Sorting Sequences, ICCV2017

[3] Self-Supervised Video Representation Learning With Odd-One-Out Networks, CVPR2017

[4] VideoChat-R1: Enhancing Spatio-Temporal Perception via Reinforcement Fine-Tuning, arxiv 2025

---

> ### Author Rebuttal · Authors · 2025-07-30
>
> > Q1: The temporal reward is mainly based on frame order. The order-relevant supervision has been widely investigated in the video undestanding community such as [1-3].
>
> A1:  Thank you for pointing this out. Prior works have indeed explored temporal order as a form of self-supervised learning signal in video understanding tasks [1–3], such as predicting or verifying frame order. In contrast, our work investigates **how to leverage temporal order as a reward signal within a GRPO-based RL framework to improve video reasoning in MLLMs**. We will cite these related works and include a discussion of them in the final version.
>
> [1] Shuffle and Learn: Unsupervised Learning using Temporal Order Verification, ECCV2016
>
> [2] Unsupervised Representation Learning by Sorting Sequences, ICCV2017
>
> [3] Self-Supervised Video Representation Learning With Odd-One-Out Networks, CVPR2017
>
> > Q2：The evalution is mainly based on video question answering. There are more temporal-relavent tasks such as temporal grounding and object tracking, which should be further investigated like VideoChat-R1
>
> A2：Thanks for your comments. As the first work to explore R1-style reinforcement learning for video reasoning in MLLMs, we primarily validate our method on video question answering, similar to prior work such as Vision-R1 [1]. In principle, our framework can also be extended to other temporally grounded tasks such as temporal grounding or object tracking. Due to the limited rebuttal time, we will investigate these tasks in the future to build a more general video reasoning model.
>
> [1] Vision-R1: Incentivizing Reasoning Capability in Multimodal Large Language Models
>
>
>
> > Q3：Small Concern: What is Qwen2.5-VL-7B (CoT)? Moreover, Qwen2.5-VL-7B (CoT) is generally worse than Qwen2.5-VL-7B-SFT on video reasoning benchmarks, while it is often better than Qwen2.5-VL-7B-SFT on video general benchmarks such as TempCompass and VideoMME. Why this happens?
>
> A3: Thanks for your comments. Qwen2.5-VL-7B (CoT) refers to the Qwen2.5-VL-7B-Instruct model directly prompted to generate COT reasoning responses without any additional training. In contrast, Qwen2.5-VL-7B-SFT is the same model fine-tuned on our curated Video-R1-CoT-165k dataset using SFT.
>
> Interestingly, we observe that Qwen2.5-VL-7B-SFT performs worse than Qwen2.5-VL-7B (CoT) on some general video benchmarks such as TempCompass and VideoMME. This highlights the limitations of SFT, which tends to rely on memorization and may generalize poorly to unseen test data [1]. In contrast, after RL training with our GRPO-based method, the model achieves consistent improvements across all benchmarks, including both reasoning and general understanding tasks. This aligns with the conclusion from prior work [1]: *“SFT memorizes, RL generalizes.”*
>
> However, SFT still plays a critical role as a cold-start phase. Our ablation study in Table 2 of the manuscript shows that removing this phase (i.e., in Video-R1-zero, which skips SFT and directly applies RL) leads to inferior performance compared to the full SFT+RL pipeline.
>
> [1]  SFT Memorizes, RL Generalizes: A Comparative Study of Foundation Model Post-training

---

> > ### Comment · Reviewer_oq4J · 2025-08-04
> >
> > Thanks for the rebuttal. The authors have addressed most of my main concerns.

---

> > > ### Author Response · Authors · 2025-08-05
> > >
> > > Thank you for the feedback. We're glad to hear that most of your concerns have been successfully addressed.

---

### Official Review · Reviewer_d9TR · 2025-07-03

**Clarity:** 3
**Significance:** 2
**Originality:** 3
**Rating:** 4
**Confidence:** 5

**Summary:**

This paper proposes a variant of GRPO for video understanding, enhancing temporal reasoning. The authors claim that existing multi-modal large language models do not fully utilize the temporal information. To address this, the authors introduced Video-R1-260K and Video-R1-CoT-165K curated datasets. Following DeepSeek-R1, the authors used rule-based rewards depending on data types. As presented in Figure 3, the variant of GRPO, T-GRPO, utilized randomly shuffled videos and their outputs. Based on the relation between responses from original and shuffled videos, rewards are differently augmented. Also, if the length of the response is reasonable, a higher reward is given. This simple variant provides considerable performance gain.

**Questions:**

1. Are the responses to original and shuffled videos paired?
2. How did you curate newly introduced datasets?
3. How sensitive is the proposed method to the additional reward values for temporal reasoning, and length in (2) and (5)?
4. Regarding temporally augmented reward defined in (2) why r_i  is given when the response is not correct?
5. Is the proposed method applicable to other temporal data? applications?

**Ethical Concerns:**

["NO or VERY MINOR ethics concerns only"]

**Final Justification:**

My all major questions are answered and concerns are addressed by rebuttal except for the limited contribution. I read other reviewers comments and I believe that this work is timely developed to improve the generalization power on temporal data. This work will provide useful insight to improve alignment methods with various data augmentation.

**Limitations:**

Yes.

**Paper Formatting Concerns:**

No formatting concern.

**Quality:**

2

**Strengths And Weaknesses:**

**Strengths**

1. **Simplicity.** The proposed method is a simple modification of the GRPO using data augmentation and modified rule-based reward functions with additional rewards when the rate of correct responses of the original data is higher than shuffled videos. Also, the responses have reasonably lengths, the authors give a higher reward. This is extremely simple and applicable to various criteria.
2. **Motivation.** After the advance of DeepSeekMath trained with GRPO, numerous papers have studied variants of GRPO. In MLLMs, or LLM-based agents, post-training became important to achieve competitive performance in downstream applications. Especially, the

**Weaknesses**

1. **Limited technical contribution.** The simplicity of solutions is desirable but no theoretical analysis or derivation is provided. Also, the relation between the proposed method and other alternatives are not discussed. GRPO is relatively new but many variants of GRPO have been studied. Without comparing the proposed method with existing GRPO variants and discussing the difference, it is difficult to the novelty of this work.
2. **Hyperparameters.** The manually designed reward functions and additional rewards entail hyperparameters. Presumably, this work is sensitive to those hyperparameters. No in-depth analysis is provided and large dependency on hyperparameters raises questions on generalizability of this work to a new data or model.
3. **Curated data.** This work requires another non-trivial manual effort to construct the dataset. The authors provided Video-R1-260K and Video-R1-CoT-165K datasets. These datasets are curated. To this extent, this work improves the performance with more annotations. As Table 1 shows mixed results. Qwen2.5-V(CoT) is better than SFT in two Video General Benchmarks: TempoCompass and  VideoMME (wo sub). However for other benchmarks, it is opposite.

---

> ### Author Rebuttal · Authors · 2025-07-30
>
> > Q1：**Limited technical contribution.** The simplicity of solutions is desirable but no theoretical analysis or derivation is provided. Also, the relation between the proposed method and other alternatives are not discussed. GRPO is relatively new but many variants of GRPO have been studied. Without comparing the proposed method with existing GRPO variants and discussing the difference, it is difficult to the novelty of this work.
>
> A1：Thanks for your comments. Several contemporary works have also explored variants of GRPO, either in general domains [1,2] or in specific domains such as agents [3]. For example, DAPO [1] introduces a clip-higher strategy and a dynamic sampling policy to improve LLM reasoning performance. GiGPO [3] proposes a two-level structure (episode-level and step-level) for estimating relative advantages to facilitate more effective RL training for agents.
>
> Different from them, we are **the first to explore a tailored GRPO algorithm for video reasoning**, in order to encourage deep temporal reasoning instead of relying on superficial visual patterns. Our ablation studies and temporal reward analysis experiments confirm the effectiveness of the proposed method. We leave more in-depth theoretical analysis for future work.
>
> To further validate the effectiveness of our method, we compare our T-GRPO with DAPO. The results are as follows:
>
> |                     | Frames | VSI-Bench | VideoMMMU | MMVU(mc) | MVBench | TempCompass | VideoMME (wo sub) |
> | ------------------- | ------ | --------- | --------- | -------- | ------- | ----------- | ----------------- |
> | Video-R1 (DAPO [1]) | 16     | 31.7      | 49.0      | 62.9     | 61.6    | 69.1        | 56.3              |
> | Video-R1 (T-GRPO)   | 16     | 34.6      | 49.8      | 64.2     | 62.7    | 72.6        | 57.4              |
>
> From the above table, it can be observed that the model trained by our proposed T-GRPO obtains better performance than DAPO. This further confirms the effectiveness of our method. We will add this discussion in the next version.
>
> [1] DAPO: An Open-Source LLM Reinforcement Learning System at Scale
>
> [2] Beyond the 80/20 Rule: High-Entropy Minority Tokens Drive Effective Reinforcement Learning for LLM Reasoning
>
> [3] Group-in-Group Policy Optimization for LLM Agent Training
>
>
>
> > Q2：**Hyperparameters.** The manually designed reward functions and additional rewards entail hyperparameters. Presumably, this work is sensitive to those hyperparameters. No in-depth analysis is provided and large dependency on hyperparameters raises questions on generalizability of this work to a new data or model.
> >
> > How sensitive is the proposed method to the additional reward values for temporal reasoning, and length in (2) and (5)?
>
> A2: Thanks for your comments. While large-scale sensitivity analysis can be computationally intensive for MLLMs, **we have already included an ablation study on the temporal reward in Eq. (2), as shown in Section 4.5 of our submitted manuscript.** The relevant information is as follows:
>
> | Temporal Reward                     | $\alpha=0.1$ | $\alpha=0.2$ | $\alpha=0.3$ | $\alpha=0.4$ |
> | ----------------------------------- | ------------ | ------------ | ------------ | ------------ |
> | Average Performance on 6 benchmarks | 55.4         | 56.8         | 56.9         | 55.1         |
>
> We find that performance slightly declines when $\alpha=0.1$ or $\alpha=0.4$, whereas $\alpha=0.2$ and $\alpha=0.3$ result in consistently strong performance. This suggests that the model is robust to the choice of $\alpha$ within a moderate range.
>
> Besides, we also add a sensitivity analysis of the length in Eq.(5). The results are as follows.
>
> | Min Length                          | $l_{min}=256$ | $l_{min}=320$ | $l_{min}=384$ |
> | ----------------------------------- | ------------- | ------------- | ------------- |
> | Average Performance on 6 benchmarks | 56.7          | 56.9          | 56.4          |
> | **Max Length**                      | $l_{max}=448$ | $l_{max}=512$ | $l_{max}=576$ |
> | Average Performance on 6 benchmarks | 56.5          | 56.9          | 56.8          |
>
> From the above table, we can see that our method is also insensitive to the choice of $l_{min}$ and $l_{max}$. However, completely removing the length reward leads to performance degradation, as discussed in Appendix A.2 of our manuscript.
>
>
>
> > Q3：**Curated data.** This work requires another non-trivial manual effort to construct the dataset. The authors provided Video-R1-260K and Video-R1-CoT-165K datasets. These datasets are curated. To this extent, this work improves the performance with more annotations. As Table 1 shows mixed results. Qwen2.5-V(CoT) is better than SFT in two Video General Benchmarks: TempoCompass and VideoMME (wo sub). However for other benchmarks, it is opposite.
>
> A3：Thanks for your comments. Interestingly, we observe that Qwen2.5-VL-7B-SFT performs worse than Qwen2.5-VL-7B (CoT) in some cases. This highlights the limitations of SFT, which tends to rely on memorization and may generalize poorly to unseen test data [1]. In contrast, after RL training with our GRPO-based method, the model achieves consistent improvements across all benchmarks, including both reasoning and general understanding tasks. This aligns with the conclusion from prior work [1]: *“SFT memorizes, RL generalizes.”*
>
> However, SFT still plays a critical role as a cold-start phase. Our ablation study in Table 2 of the manuscript shows that removing this phase (i.e., in Video-R1-zero, which skips SFT and directly applies RL) leads to inferior performance compared to the full SFT+RL pipeline. This highlights the value of our curated Video-R1-CoT-165k dataset. We hope our datasets can serve as a useful resource for future research in the community.
>
> [1]  SFT Memorizes, RL Generalizes: A Comparative Study of Foundation Model Post-training
>
>
>
> > Q4： Are the responses to original and shuffled videos paired?
>
> A4: Thanks for your comments. They are not paired. We respectively sample responses from the original and shuffled videos. If the proportion of correct answers in the original group is higher than that in the shuffled group, a positive temporal reward is assigned.
>
>
>
> > Q5：How did you curate newly introduced datasets?
>
> A5: Thanks for your comments. For dataset construction, we incorporated both image-based and video-based reasoning data. The image-based data was collected from a range of public datasets and selected to cover diverse domains (e.g., mathematics, spatial reasoning, and expert-level knowledge) and varying difficulty levels. Its purpose is to help the model acquire general reasoning skills in static contexts. For video-based data, our focus is on enabling temporal reasoning abilities, such as understanding event progression, motion dynamics, and frame-to-frame dependencies. We collect data from a variety of public datasets and carefully sample and balance the proportion of each subset. All samples in Video-R1-260k are standardized to the same format for consistency.
>
> To facilitate an effective SFT cold start, we leveraged Qwen2.5-VL-72B-Instruct to generate Chain-of-Thought (CoT) rationales for the Video-R1-260k samples using a structured prompt (see Appendix C.1). After applying rule-based filtering to eliminate low-quality or inconsistent generations, we obtained a high-quality subset—Video-R1-CoT-165k—which we used during the initial supervised training stage.
>
> We believe our carefully curated datasets, combining broad reasoning coverage and temporal reasoning supervision, provides a strong foundation for training multimodal LLMs on video reasoning tasks.
>
>
>
> > Q6：Regarding temporally augmented reward defined in (2) why r_i is given when the response is not correct?
>
> A6：Thanks for your comments. We follow the original GRPO reward design in Deepseek-R1, where $r_i$ includes both correctness reward and format reward. If the answer is incorrect but the format is valid (or vice versa), the reward $r_i$ is computed as 0 + 1 = 1. If both the answer and format are incorrect, the reward $r_i$ is 0 + 0 = 0. If both are correct, the reward $r_i$ is 1 + 1 = 2. We will clarify this in the final version.
>
>
>
> > Q7：Is the proposed method applicable to other temporal data? applications?
>
> A7: Thanks for your comments. In principle, our proposed method can be applied to other temporal tasks. We leave broader applications for future exploration.

---

### Official Review · Reviewer_Exe4 · 2025-07-08

**Clarity:** 3
**Significance:** 3
**Originality:** 3
**Rating:** 5
**Confidence:** 5

**Summary:**

This paper introduces Video-R1, a framework that systematically explores the R1 paradigm to enhance video reasoning in large multimodal models. The authors identify two main challenges when applying the GRPO algorithm to video reasoning: (1) insufficient reward for temporal reasoning, which can lead to shortcut solutions, and (2) a lack of high-quality video reasoning data. To address these, they propose the T-GRPO algorithm, which incentivizes the use of temporal information by rewarding the model only when it performs better on temporally ordered frames compared to shuffled ones. Additionally, they construct two new datasets: Video-R1-CoT-165k for SFT and Video-R1-260k for RL. Experimental results show that Video-R1 significantly improves performance on both video reasoning and general video benchmarks.

**Questions:**

Please refer to weakness part.

**Ethical Concerns:**

["NO or VERY MINOR ethics concerns only"]

**Final Justification:**

I am satisfied with the authors’ response, which addresses my main concerns. Video-R1 represents an early exploration of video RLVR, with contributions in performance, data, and methodology. Accordingly, I have increased my rating.

**Limitations:**

yes

**Quality:**

3

**Strengths And Weaknesses:**

### Strengths

Video-R1 is a representative early exploration of video reasoning with large multimodal models (open-source), distinguished by its strong performance. In addition to the benchmark results reported in the paper, Video-R1 also achieves top performance on recent released benchmark [a]. The proposed T-GRPO algorithm is both simple and effective, encouraging the model to reason more deeply about temporal relationships in video data.

[a] Video-Holmes: Can MLLM Think like Holmes for Complex Video Reasoning? arXiv: 2505.21374.

### Weaknesses

- 1. It seems that the paper does not report the original Qwen2.5-VL-Instruct performance. Please include this baseline using the same inference parameters as your model. Qwen2.5-VL-7B-Instruct should also be competitive. This comparison is essential for readers to clearly understand the benefits of your approach.

- 2. Given access to 8x80G GPUs, why not train on longer sequences? With Qwen-style smart resize, the model should support at least 24k sequence length (approximately 240 frames) during training. To convincingly demonstrate improvements in temporal reasoning, please train and evaluate the model on more frames and provide comparative results.

- 3. Concerns with T-GRPO Algorithm Setting:
    - a. Since training is limited to 16 frames, longer videos (e.g., >2 minutes. BTW, please specify the video durations in Video-R1-CoT-165k and Video-R1-260k) will lose significant temporal correlation. This may make the distinction between temporally ordered and randomly shuffled frames ambiguous, potentially weakening the proposed temporal reward.
    - b. The rationale for why temporally ordered frames should yield higher rewards is unclear. If the model can reason correctly from randomly shuffled frames, does this not indicate even stronger temporal reasoning, as it can "recover" the original order?
    - c. While most video LLMs use causal visual token order, some employ full-attention encoders, making it difficult to guarantee that token order reflects true temporal order. Your approach is based on Qwen2.5VL uses a single-frame encoder, ensuring visual order matches temporal causality. Please clarify that in your methods section.

---

> ### Author Rebuttal · Authors · 2025-07-30
>
> > Q1: It seems that the paper does not report the original Qwen2.5-VL-Instruct performance. Please include this baseline using the same inference parameters as your model. Qwen2.5-VL-7B-Instruct should also be competitive. This comparison is essential for readers to clearly understand the benefits of your approach.
>
> A1: Thanks for your advice. We include this baseline for comparison as follows:
>
> |                        | Frames | VSI-Bench | VideoMMMU | MMVU(mc) | MVBench | TempCompass | VideoMME (wo sub) |
> | ---------------------- | ------ | --------- | --------- | -------- | ------- | ----------- | ----------------- |
> | Qwen2.5-VL-7B-Instruct | 16     | 30.2      | 47.2      | 61.9     | 60.2    | 71.6        | 55.8              |
> | Video-R1               | 16     | 34.6      | 49.8      | 64.2     | 62.7    | 72.6        | 57.4              |
>
> From the above table, we can find that our Video-R1 still outperforms Qwen2.5-VL-Instruct, illustrating the effectiveness of our method. We will add this baseline in the next version.
>
>
>
> > Q2：Given access to 8x80G GPUs, why not train on longer sequences? With Qwen-style smart resize, the model should support at least 24k sequence length (approximately 240 frames) during training. To convincingly demonstrate improvements in temporal reasoning, please train and evaluate the model on more frames and provide comparative results.
>
> A2：Thanks for your comments. Unlike standard SFT, training models with GRPO demands significantly more computational resources due to several factors: 1. Multiple sampling per question (usually more than 8) 2. Loading a reference model $\pi_{ref}$ with the same size as the reasoning model for calculating the KL divergence 3. Long COT output (700~1000 tokens for some questions) . As a result, we encounter **out-of-memory (OOM) issues when training with 64 or more frames** on 8×80GB GPUs.
>
> To further demonstrate our method’s effectiveness with longer sequences, we have trained the model on 48 frames and evaluated it on 64 and 128 frames. The results are as follows:
>
> |          | Train Frames | Evaluation Frames | VSI-Bench | VideoMMMU | MMVU(mc) | MVBench | TempCompass | VideoMME (wo sub) |
> | -------- | ------------ | ----------------- | --------- | --------- | -------- | ------- | ----------- | ----------------- |
> | Video-R1 | 16           | 64                | 37.1      | 52.4      | 63.8     | 64.8    | 73.2        | 61.4              |
> | Video-R1 | 48           | 64                | 37.6      | 52.8      | 65.0     | 65.3    | 74.0        | 62.9              |
> | Video-R1 | 48           | 128               | 38.3      | 52.8      | 66.2     | 65.7    | 74.2        | 63.8              |
>
> We can find that training and evaluating on more frames generally yields better performance. We will explore experiments on more frames in the future.
>
>
>
> > Q3：Since training is limited to 16 frames, longer videos (e.g., >2 minutes. BTW, please specify the video durations in Video-R1-CoT-165k and Video-R1-260k) will lose significant temporal correlation. This may make the distinction between temporally ordered and randomly shuffled frames ambiguous, potentially weakening the proposed temporal reward.
>
> A3: Thanks for your comments. We provide the duration statistics of our datasets in the following table:
>
> |                   | Average duration | Percentage of Videos < 2 min |
> | ----------------- | ---------------- | ---------------------------- |
> | Video-R1-CoT-165k | 38.78 s          | 94.0%                        |
> | Video-R1-260k     | 36.87 s          | 94.9%                        |
>
> As shown, the average video length in both datasets is approximately 40 seconds, and over 94% of the videos are shorter than 2 minutes. This suggests that sampling 16 frames still allows the model to capture a reasonable portion of the temporal structure in most cases. We agree that incorporating more frames for  continuous temporal modeling could further improve performance for long video, and we consider this a promising direction for future work.
>
>
>
> > Q4： The rationale for why temporally ordered frames should yield higher rewards is unclear. If the model can reason correctly from randomly shuffled frames, does this not indicate even stronger temporal reasoning, as it can "recover" the original order?
>
> A4：Thanks for your comments. In practice, the model usually tends to take shortcuts for reasoning rather than trying to employ/recover the temporal order. Specifically, it often relies on spurious cues, such as a single frame or snapshot, to guess the answer, while neglecting the overall temporal information. We observe that the model can sometimes arrive at the correct answer through such incorrect reasoning paths. Under GRPO, these paths are rewarded equally, which can lead to suboptimal policies and poor generalization.
>
> Therefore, we explicitly design the reward signal to favor trajectories derived from temporally ordered inputs, encouraging the model to learn policies that capture temporal structures rather than rely on spurious cues. One contemporary work [1] in July 2025 also adopts a similar approach in image reasoning domain, which contrasts the response probabilities between normal image input and corrupted image input.
>
>
>
> [1] Perception-Aware Policy Optimization for Multimodal Reasoning
>
>
>
> > Q5： While most video LLMs use causal visual token order, some employ full-attention encoders, making it difficult to guarantee that token order reflects true temporal order. Your approach is based on Qwen2.5VL uses a single-frame encoder, ensuring visual order matches temporal causality. Please clarify that in your methods section.
>
> A5: Thanks for your advice. We will clarify this in the paper.

---

> > ### Comment · Reviewer_Exe4 · 2025-08-05
> >
> > Thanks for the response. My concern has been addressed. One more thing: do you have any thoughts on applying T-GRPO with the LongVILA MR-SP [1] method?
> >
> > [1] Scaling RL to Long Videos. arXiv:2507.07966v3.

---

> ### Author Response · Authors · 2025-08-05
>
> Thank you for the follow-up and we're glad to hear that your concerns have been addressed.
>
> MR-SP [1] provides a scalable and memory-efficient backbone for reinforcement learning on long videos, enabling training setups that would otherwise be infeasible. Its design naturally supports integrating other algorithms like T-GRPO.
>
> In Stage 1 (Rollout with Paralleled Encoding), MR-SP leverages sequence parallelism to encode long videos by dividing frames across multiple GPUs. To support T-GRPO, we can extend this mechanism to process both temporally ordered and shuffled versions of each video in parallel, caching both sets of embeddings for reuse across rollouts.
>
> In Stage 2 (Prefilling with Sequence Parallelism), the gathered embeddings from both temporally ordered and shuffled versions can be fed into the policy and reference models using MR-SP’s parallel prefilling pipeline. Each version is sharded across devices with uniform padding, allowing logits to be computed in parallel without additional overhead.
>
> In summary, MR-SP's two-stage pipeline could align with T-GRPO's training. The combination enables scalable, temporally aware reinforcement learning on long videos, and we are excited to explore this integration in future work.

---

### Decision · Program_Chairs · 2025-09-17

**Decision:**

Accept (poster)

**Comment:**

This paper proposes an R1-style training method for video MLLMs, adapting the GRPO algorithm to enhance temporal-based video reasoning.

The reviewers are generally positive about the work, acknowledging that: 1) R1-style training for video MLLMs is a timely and important topic, 2) the approach yields noticeable gains across multiple benchmarks, and 3) the paper is generally well-written and easy to follow.

The authors' rebuttal successfully addressed the few initial concerns raised by the reviewers. After carefully reading the reviews, the rebuttal, and all discussions, this meta-reviewer finds no reason to overturn the reviewers' unanimous recommendation. This paper is a clear accept.